# `IAFormer`: Interaction-Aware Transformer network for collider data analysis

W. Esmail[a], A. Hammad[b] and M. Nojiri[b,c,d]

[a] Institut für Kernphysik, Universität Münster, Wilhelm-Klemm-Str. 9, 48149 Münster, Germany.
[b] Theory Center, IPNS, KEK, 1-1 Oho, Tsukuba, Ibaraki 305-0801, Japan.
[c] The Graduate University of Advanced Studies (Sokendai), 1-1 Oho, Tsukuba, Japan.
[d] Kavli IPMU (WPI), University of Tokyo, 5-1-5 Kashiwanoha, Kashiwa, Chiba 277-8583, Japan.

**Abstract**

In this paper, we introduce `IAFormer`, a novel Transformer-based architecture that efficiently integrates pairwise particle interactions through a dynamic sparse attention mechanism. `IAFormer` has two new mechanisms within the model. First, the attention matrix depends on predefined boost invariant pairwise quantities, reducing the network parameters significantly from the original particle transformer models. Second, `IAFormer` incorporates the sparse attention mechanism by utilizing the "differential attention", so that it can dynamically prioritize relevant particle tokens while reducing computational overhead associated with less informative ones. This approach significantly lowers the model complexity without compromising performance. Despite being computationally efficient by more than an order of magnitude than the Particle Transformer network, `IAFormer` achieves state-of-the-art performance in classification tasks on the top and quark-gluon datasets. Furthermore, we employ AI interpretability techniques, verifying that the model effectively captures physically meaningful information layer by layer through its sparse attention mechanism, building an efficient network output that is resistant to statistical fluctuations. `IAFormer` highlights the need for sparse attention in Transformer analysis to reduce the network size while improving its performance.

# 1 Introduction

Jets are fundamental objects for LHC collision data, and choosing an appropriate jet tagging algorithm is essential to uncover intriguing physics phenomena. For scenarios involving the production of a high momentum, hadronically decaying heavy particle, such as W and Z bosons or top quarks, a large radius jet is often used to capture entire particles arising from the heavy particle decay.

The recognition of fat jets with the multi-prong structure was first noted in Ref. [1], where jet substructure techniques emerged as a powerful method for suppressing QCD backgrounds, thereby increasing the sensitivity of Beyond Standard Model (BSM) scenarios at the LHC. The pronged structure associated with different heavy resonant particles is dictated by the final state particles of their decays. Over time, jet substructure methodologies have evolved, leading to Deep Learning (DL) based classification techniques for tagging

heavy particles [2–21]. These approaches are now widely applied in LHC analyses conducted by ATLAS and CMS [22–27]. They are also employed in scenarios with multiple heavy particle final states [28–30] and in extracting fundamental parameters of the theory [31–35]. In contrast, processes that generate spatially isolated quarks or gluons are typically reconstructed using one or more small radius jets. This is followed by flavour tagging, a technique that identifies the initiating parton, whether an up, down, strange, charm, bottom quark, or a gluon. Bottom and charm hadrons (primarily B and D hadrons) have relatively long lifetimes, allowing them to travel a short distance within the detector before decaying. Their decays produce displaced tracks originating from a secondary vertex, distinct from the primary collision point. By leveraging machine-learning techniques, this vertex displacement information, combined with subsequent decay properties, significantly improves tagging efficiency [36–39]. Additionally, studies have explored the potential for strange-quark identification using ML techniques [40–42]. Since gluons belong to the octet representation of QCD SU(3) symmetry, their jet characteristics differ from those of quarks, which are in the triplet representation. Consequently, ML-based methods have also been developed to distinguish gluon-initiated jets from quark-initiated ones [43–48].

Over the past few decades, significant advancements have been made in jet identification techniques at hadron colliders. In particular, DL methods have transformed how we analyze and utilize jet-related observables. Treating a jet as a single entity provides limited information about its origin, whereas a more detailed analysis of its internal structure and constituent particles offers valuable insights into the nature of the initiating particle. This has led to the development of various techniques collectively referred to as jet tagging, which aim to classify jets based on their underlying physics.

Recently, Transformer networks have demonstrated significant improvements in jet tagging tasks[21, 49, 50]. Standard Transformers process jet constituents along with their corresponding features, which include kinematic information for each hadron inside the jet. In this setup, the network assigns weights to each particle based on its feature importance in the overall classification decision.

Incorporating features derived from particle-pair interactions further enhances network performance, as these features reflect the radiation patterns inside the jet [10]. The pairwise interactions have been integrated into the attention mechanism either using an interaction matrix as a bias for the computed attention scores [49] or replacing the Query and Key matrices with the interaction matrix itself [50]. In both approaches, the number of attention heads must be equal to or larger than the number of features in the interaction matrix. Additionally, the interaction matrix remains fixed across attention heads and is injected into each attention layer without updates. These rigid structure requirements can disrupt the learning of attention patterns across different layers, potentially limiting the overall effectiveness. `IAFormer` addresses these challenges by allowing the interaction matrix to be updated across different attention heads and Transformer layers. Furthermore, the interaction matrix is refined through skip connections and propagated to later attention layers.

Another key innovation in `IAFormer` lies in its ability to dynamically focus on the most important particles while suppressing attention to less relevant ones. This is achieved through a dynamic sparse attention mechanism, implemented as the difference between two copies of the learned interaction matrix [51]. By encouraging the network to learn distinct representations for these copies, `IAFormer` prioritizes essential hadrons while de-emphasizing soft radiation, which often exhibits similar patterns in both signal and background.

`IAFormer` architecture demonstrates superior performance compared to previously introduced Transformer-based models when evaluated on top tagging and quark-gluon tagging

tasks with a significantly reduced network size. Moreover, `IAFormer` is designed as a general framework that can be adapted for various classification tasks. The implementation is publicly available at IAFormer GitHub, and detailed usage instructions are provided in the appendix.

This paper is organized as follows: Section 2 discusses various Transformer network architectures, including the structure of `IAFormer`. Section 3 presents validation results, comparing `IAFormer` with existing networks for top tagging and quark-gluon tagging tasks. In Section 4, we employ different interpretability methods to analyze the learned attention patterns in `IAFormer` during the top tagging task. Section 5 provides a discussion and conclusion.

# 2 Transformers for jet tagging

Transformers were originally introduced for natural language processing, particularly for sequence-to-sequence tasks like machine translation [52], which were later modified for collider physics. The core of Transformer is the self-attention mechanism, which dynamically assigns importance to different input components. One of the most significant advantages of self-attention is its ability to capture long-range dependencies, as it considers all elements of the input simultaneously rather than relying on sequential processing. Moreover, modifications to the self-attention formula by incorporating the particle pair interactions enhance the network's performance.

In jet tagging tasks, the outcome of an event should remain unchanged regardless of the order in which particles are processed. Therefore, the designed DL networks should treat particles as an unordered set, commonly referred to as a "particle cloud". By representing hardons inside the jet in a permutation-invariant manner, particle cloud models overcome the challenges posed by combinatorial ambiguities. The particle-based methods also provide the flexibility to incorporate additional particle attributes, such as particle identification and vertex information.

Several machine learning architectures have been developed for jet tagging under the permutation invariant framework, including Deep Sets [53], Edge Convolution Networks [54], and Transformers [16, 20, 29, 49, 55, 56]. Deep Sets can achieve state-of-the-art performance but require a large latent space, making them computationally demanding. Edge Convolution Neural Networks (EDGCNN) address this challenge by leveraging local neighborhood information within the particle cloud to enhance learning. For the case of particle transformers, attention weights are computed for every particle interaction within the dataset, and outputs are aggregated to keep the permutation invariance, so that the network output remains unaffected by the order of the input. However, self-attention comes with computational challenges, particularly its quadratic scaling with input size. Various improvements, such as sparse attention [57], have been proposed to address this issue.

In the following, we first introduce the Plain Transformer and its modified version, including the interaction matrix, then present the new network, `IAFormer`. `IAFormer` significantly reduces the computational burden of attention matrix calculation by incorporating prefixed pairwise information with trainable weights. Moreover, it reduces statistical fluctuations by introducing a dynamic spatial attention mechanism, referred to as differential attention.

## 2.1 Plain Transformer

The core component of a transformer model is the attention matrix, a weighted sum of the input elements. The weights are determined based on the relevance of each element to the

others. For the particle transformer, the attention weights are computed in a permutation-invariant manner, and the resulting model is able to process sets efficiently.

Mathematically, the attention mechanism operates as follows: The input to the multi-head attention $X$ has a grid-like structure with dimensions $I \times J$, with $I$ indicating the number of particles and $J$ indicating the relevant features for each particle in the dataset.

For given $X_{ij}$, the feature vector $X_{*j}$ is embedded into a higher dimension via an MLP. The embedding to a higher dimension helps the network to capture meaningful complex patterns and nonlinear relationships that may not be apparent in the original feature space; therefore, it enhances the model's ability to distinguish different classes for deep learning tasks like classification and anomaly detection. High-dimensional embeddings also provide richer representations that facilitate downstream tasks such as contrastive learning and clustering. The size of the embedded features is determined by the number of neurons in the final MLP layer, as

$$X_{i,j}^{\text{embed},(l)} = \sigma \left( X_{i,j'}^{(l-1)} W^{(l)}{}_{j',j} + b^{(l)}{}_j \right) , \tag{1}$$

with $l$ is the number of the MLP layer, $\sigma$ is a nonlinear activation function, and $W$ and $b$ are the layer weights and bias.

Once the input features are mapped to a higher dimension, the inputs are passed to three separate linear layers, forming the Query, Key, and Value matrices as

$$Q_{i,j}^{I \times J} \equiv X_{i,j'}^{\text{embed}} \cdot W_{j',j}^Q , \quad K_{i,j}^{I \times J} \equiv X_{i,j'}^{\text{embed}} \cdot W_{j',j}^K , \quad V_{i,j}^{I \times J} \equiv X_{i,j'}^{\text{embed}} \cdot W_{j',j}^V , \tag{2}$$

where $Q$, $K$, and $V$ are the query, key, and value matrices, respectively, used to compute the attention for the dataset. The suffix $I \times J$ indicates the dimension of the matrices, where $I$ is the number of particle tokens and $J$ is the number of embedded features. The weight matrices have the dimensions of $I \times I$ to preserve the dimensions of the embedded input dataset. The attention score $\alpha$ is defined as:

$$\alpha^{I \times I} \equiv \text{softmax} \left( \frac{Q \cdot K^T}{\sqrt{d}} \right) = \frac{\exp(Q \cdot (K)^T / \sqrt{d})}{\sum \exp(Q \cdot (K)^T / \sqrt{d})} \tag{3}$$

where suffix $I \times I$ of $\alpha$ indicates the dimension of the attention weight matrix $\alpha$. In the last line, the sum runs over all components $I \times I$, and $d$ is the length of the particle tokens.

The attention score represents the relative importance of one element in a sequence with respect to another element in the context of a given task. The definition of $\alpha$ with softmax on the scaled dot product ensures that the attention distributed across different elements sums to one. This self-attention mechanism determines how much focus should be given to different input elements when making predictions. Unlike traditional sequence models, which rely on predefined relations among the inputs, attention mechanisms dynamically assign different weights to different parts of the input, allowing the model to emphasize the most relevant features without prejudice.

The attention output is computed as

$$\mathcal{Z}_{i,j} = \alpha_{i,i'} \cdot V_{i',j} . \tag{4}$$

The attention output matrix has the same dimensions as the first input dataset, $X^{I \times J}$. Each transformer layer incorporates a multi-head attention mechanism, which combines attention heads to enable parallel and multi-dimensional processing of inputs. This approach enhances the network performance as multi-head attention increases the expressiveness of the model. Each head learns its own projection of the input data, which means the model can explore

different subspaces of the feature space. This is particularly useful for capturing both local and global dependencies within the input. As the input dataset is structured in an unordered way, for example, some heads may focus on short-range dependencies, with low $\Delta R$, while others may capture long-range relationships, with larger $\Delta R$.

The outputs from the different attention heads are linearly combined to produce the final output of the multi-head attention layer. The output of the multi-head cross-attention is given by

$$\mathcal{O}^{I \times J} = \text{concat} \left( \mathcal{Z}^1_{I \times J}, \mathcal{Z}^2_{I \times J}, \ldots, \mathcal{Z}^n_{I \times J} \right) \cdot W_{n \cdot J \times J}, \tag{5}$$

with $W_{n \cdot J \times J}$ is the learnable linear transformation matrix, so that the final output $\mathcal{O}$ maintains the dimensions of the input dataset. When the outputs of different attention heads are concatenated and projected back into the model feature space, the model effectively combines multiple views of the data into a unified representation. This fusion of different perspectives contributes to the robustness and flexibility of Transformer models.

Finally, the attention output is used to scale the input dataset via a skip connection

$$\widetilde{X}^{I \times J} = X^{I \times J} + \mathcal{O}^{I \times J}. \tag{6}$$

The transformed dataset, $\widetilde{X}$, represents the relative importance of each element within the dataset, captured from all particles, to the network decision making. Moreover, the transformed dataset has the same dimensions as the input one, which enables stacking more Transformer layers for better performance.

## 2.2 Transformer with interaction matrix

The pairwise interaction matrix is the matrix of the prefixed quantities calculated from the pair of particle tokens. For the jet classification, the precomputed variables are mass, relative angles, $k_T$, etc, of the pair of jet constituents. The standard Transformer architecture does not inherently support the integration of the pairwise interaction matrix within its attention mechanism. The pairwise interaction matrix has dimensions of (particles $\times$ particles $\times$ pairwise features). A straightforward approach to incorporating the pairwise interaction matrix is by embedding it directly into the attention score, equation 3. The feature dimension is adjusted to match the number of attention heads using a linear embedding layer. Consequently, the attention score structure for a single attention head, incorporating the interaction matrix, takes the form that is utilized by ParT[49],

$$\alpha^{I \times I} = \text{softmax} \left( \frac{Q^{I \times J} \cdot (K^{I \times J})^T}{\sqrt{d}} + \mathcal{I}^{I \times I} \right), \tag{7}$$

with $\mathcal{I}_{i,j}$ is the pairwise interaction matrix for a single head after the embedding. In this structure, the interaction matrix acts as a bias in the attention score.

While this approach effectively integrates the interaction matrix into the attention score, it has certain limitation. The feature dimension of the interaction matrix needs to be reshaped to be compatible with the specified number of attention heads. This occurs because $Q$ and $K$ must accommodate the same number of attention heads in this setup.

Another approach is to replace $Q$ and $K$ matrices by the interaction matrix [50]. This modifies the attention score as

$$\alpha^{I \times I} = \text{softmax} \left( \mathcal{I}^{I \times I} \right). \tag{8}$$

Omitting the Query and Key matrices eliminates the necessity of expanding the input feature dimensions of $Q \cdot K^T$. Therefore, this structure mitigates the issue of increasing network parameters as the number of features in the pairwise interaction matrix grows.

While this approach reduces network complexity and improves performance, it has a key limitation. Because the interaction matrix is added independently at each Transformer layer, a mismatch can arise between the feature representations of the pairwise interaction matrix and the input $X_{i,j}$. Since the attention mechanism assigns weights to all particle tokens in the dataset, this misalignment can lead to attention being allocated to irrelevant particles, such as soft hadrons.

## 2.3 IAFormer

IAFormer is a transformer-based architecture, but significant modifications have been made compared to the ParT setup, as illustrated in Figure 1. Instead of computing the attention score through the multiplication of the Query and Key matrices, IAFormer applies the softmax function directly to a trainable interaction matrix of the form $W\mathcal{I}$. This interaction matrix is independently optimized for each attention head within the individual attention layers, enabling the network to dynamically learn discriminative patterns among different jets with effectively reduced size.

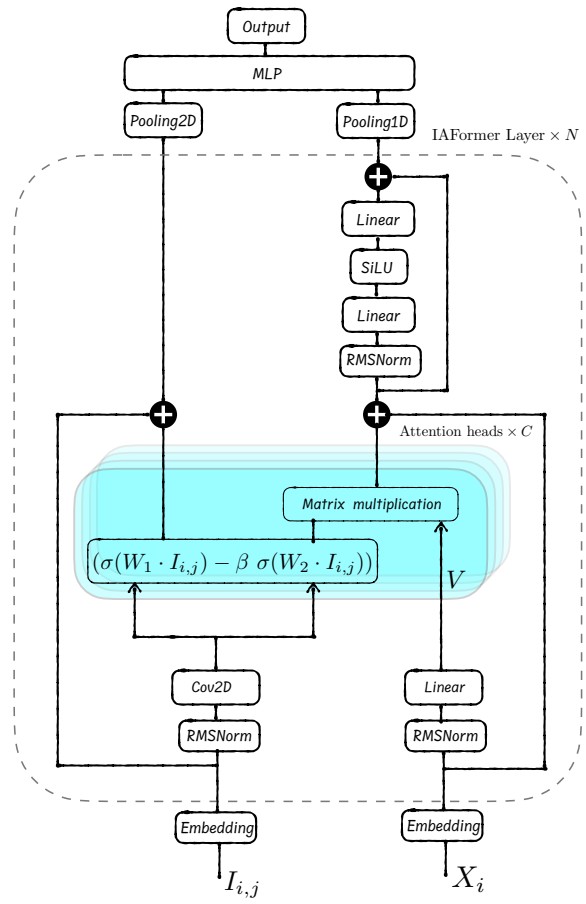

Figure 1: Schematic architecture of IAFormer network.

Additionally, a dynamic sparse attention pattern is included following the idea of "differential attention" [51]. The differential attention utilizes the attention scores as the difference between two separate softmax attention maps. The subtraction cancels noise, promoting the sparse attention patterns dynamically. The model has a trainable parameter $\beta$ to control the level of the subtraction. The parameter is initialized randomly and shared across

all attention heads within each attention layer. This mechanism suppresses attention scores for less relevant particle tokens, enhancing the model's focus on important interactions.

### 2.3.1 Dynamic sparse attention via differential attention

Sparse attention is a technique used in Transformer models to reduce the computational complexity of the self-attention mechanism by limiting the number of key-query pairs attending to the classification. Instead of computing attention over all tokens in a sequence, sparse attention selectively attends to a subset of tokens based on predefined patterns, learned structures, or efficient approximations. This reduces the memory and computational cost from quadratic to linear or sub-quadratic complexity, making it possible to process longer sequences efficiently.

Different types of sparse attention have been introduced, such as fixed pattern and learnable sparse attention. Fixed pattern sparse attention is a type of sparse attention mechanism where each token attends only to a predefined subset of tokens based on a fixed rule [58–60]. On the other hand, Learnable sparse attention dynamically determines which tokens to attend to others based on learned patterns. Learnable sparse attention allows the model to adaptively focus on the most relevant tokens in different contexts, leading to improved efficiency. In this paper, we utilize a dynamic sparse attention mechanism incorporated in the transformer called "differential attention" which was first introduced in [51],

$$\alpha_{i,i'} = \text{softmax}(W_1 \cdot \mathcal{I}_{i,j}) - \beta \, \text{softmax}(W_2 \cdot \mathcal{I}_{i,j}) \, , \tag{9}$$

with $\beta$ is a learnable vector and $W_1, W_2$ are two learnable matrices. During the training process, the value of $\beta$ vector is optimized to demote the attention score assigned to less relevant tokens, resulting in the attention score matrix to be sparse. This results in an implicit sparsity, where tokens gradually focus only on meaningful interactions while ignoring less relevant ones. The self-attention mechanism is guided by this dynamic sparsity, reducing computational overhead while preserving essential long-range dependencies. The `IAFormer` shows better performance compared with ParT using a notably smaller number of parameters.

Despite the advantages of dynamic sparse attention, a challenge remains in controlling the value of the $\beta$ parameter. In the current setup, we clip $\beta$ to lie within the range [0, 1]. Additionally, the attention score $\alpha_{i,j}$ can sometimes take negative values. In such cases, it becomes unclear how to interpret it as an attention probability.

### 2.3.2 Network structure and implementation

The `IAFormer` architecture comprises a data embedding block, attention-based layers, and a final MLP for classification, as can be seen in Figure 1. Unlike original Transformer-based models, our approach does not require a class token to aggregate learned information across the layers. Instead, we replace it with a simple pooling operation applied to the final layer output[1]. Two input datasets are used, particle kinematics with dimension $(100, 11)$ and interaction matrix with dimension $(100, 100, 6)$. Each dataset is processed through a dedicated embedding block. For the particle kinematics data, we employ an MLP consisting of three fully connected layers with 256, 128, and 32 neurons, respectively. Each layer is followed by a GELU activation function [62] and Root Mean Square Layer Normalization (RMSNorm) [63]. While different normalization techniques can be used, we found that

---

[1]The class token is particularly important in vision tasks, where an input image is divided into small patches, as it helps capture the global structure by aggregating information from Transformer layers. However, since our dataset consists of particle clouds, we replace the class token with average pooling [61].

RMSNorm enhances network convergence. For the pairwise interaction dataset, three two-dimensional convolution layers are employed with 256, 128, and 32 filters. The filter size is 1 so that the mapped features are not diluted by the convolution operation. Each convolution layer is followed by GELU activation and RMSNorm layers. The dimension of the embedded kinematic dataset is $(100, 32)$ while the embedded particle interaction dataset has the dimension $(100, 100, 32)$.

The attention-based layer has a distinct structure, processing two input datasets and outputting both datasets scaled by the computed attention. For each dataset, an RMSNorm layer is applied to normalize the input features. The trainable matrices, $W_1$ and $W_2$, are initialized using a fully connected layer for the particle kinematics dataset and a 2D convolutional layer for the pairwise interaction dataset. Both datasets are scaled using skip connections. For the particle kinematics dataset $X_i$, an MLP is added after the attention heads. This MLP consists of an RMSNorm layer followed by a fully connected layer with Sigmoid-Weighted Linear Units (SiLU) activation [64]. The MLP size is set to four times the number of embedded features. For the top tagging and quark-gluon tagging tasks, we use 12 attention layers, each containing 16 attention heads. The output of the final attention layer undergoes an average pooling before the final MLP. The final MLP layer consists of a fully connected layer with 100 neurons and an output layer with one neuron and a sigmoid activation function.

## 3  Performance of `IAFormer`

In this section, we validate `IAFormer` against existing networks for top tagging and quark-gluon classification. The validation is performed using publicly available datasets, the top [65] and the quark-gluon dataset [66]. Results for existing networks are taken from their respective references. Additionally, we compare `IAFormer` performance with two baseline networks, a Plain Transformer and a Transformer with an interaction matrix, both trained from scratch, as described in Appendix A.

### 3.1  Input variables and data structure

In this analysis, we consider two input datasets, one of which is the particle momentum features and the other is the particle pair interaction features. For the top tagging, The original data set retains up to 200 constituent 4-momenta $(px, py, pz, E)$ for each jet. For top jet tagging, we construct the input dataset with up to 100 $p_T$ ordered jet constituents with eleven features as:

$$
\begin{aligned}
&\text{- } P_4 = (p_x, p_y, p_z, E) && : \text{particle 4-momentum} \\
&\text{- } \Delta\eta = \eta - \eta_{\text{jet}} && : \text{pseudorapidity difference} \\
&\text{- } \Delta\phi = \phi - \phi_{\text{jet}} && : \text{azimuthal angle difference} \\
&\text{- } \Delta R = \sqrt{(\Delta\eta)^2 + (\Delta\phi)^2} && : \text{angular distance from jet axis} \\
&\text{- } \log(p_T) && : \text{transverse momentum (GeV)} \\
&\text{- } \log(E) && : \text{energy (GeV)} \\
&\text{- } \log\left(\frac{p_T}{p_{T_{\text{jet}}}}\right) && : \text{normalized } p_T \text{ (GeV)} \\
&\text{- } \log\left(\frac{E}{E_{\text{jet}}}\right) && : \text{normalized energy (GeV)}
\end{aligned}
\tag{10}
$$

Additionally, the quark-gluon dataset includes particle ID information, providing six extra features: the charge of each final-state particle and five binary indicators specifying whether a particle is an electron, muon, photon, charged hadron, or neutral hadron. The particle ID is represented using one-hot encoding, where a value of 1 corresponds to the given particle and 0 to all others.

For the pairwise particle interactions, we consider six common features for the top and quark gluon tagging [10]

$$
\begin{array}{lll}
\text{-} \ (p_{T_a} + p_{T_b})/p_{T_j} & : & \text{sum of pair transverse momenta normalized by jet } p_T \\
\text{-} \ (E_a + E_b)/E_j & : & \text{sum of pair energies normalized by jet energy} \\
\text{-} \ \Delta = \sqrt{(\eta_a - \eta_b)^2 + (\phi_a - \phi_b)^2} & : & \text{angular distance between particles} \\
\text{-} \ k_T = \min(p_{T_a}, p_{T_b}) \cdot \Delta & : & \text{transverse momentum scale} \\
\text{-} \ z = \min(p_{T_a}, p_{T_b})/p_{T_a} + p_{T_b} & : & \text{momentum sharing fraction} \\
\text{-} \ m^2 = (E_a + E_b)^2 - |\mathbf{p}_a + \mathbf{p}_b|^2 & : & \text{invariant mass squared of the pair}
\end{array}
\tag{11}
$$

Similar to [49], we consider the logarithm of the pairwise interaction variables. Integrating the energy and momentum variables in the interaction matrix is crucial, as `IAFormer` uses the interaction matrix only for the attention score. The first two variables identify the most energetic particle pairs that most likely form the prongs of the top jet.

The plain Transformer network uses only the particle kinematic dataset with the dimension $(100, 11)$, with the first and second numbers referring to the maximum number of jet constituents and the features, respectively. `IAFormer` uses two input datasets, particle kinematics and pairwise interaction matrix of the dimension of $(100, 100, 6)$.

Note that the interaction matrix is calculable from the particle kinematic dataset; however, in order to prioritize the inputs of the interaction matrix, one needs the knowledge of boost invariance and QCD, and a huge dataset is required for a plain Transformer to find the relevant information. The setup of `IAFormer` is therefore more efficient compared with that of Plain transformer.

## 3.2 Top tagging

The identification of jets originating from hadronically decaying top quarks, known as top tagging, plays a vital role in LHC new physics searches. To evaluate the performance of our proposed network, we employ the top tagging dataset. This dataset consists of jets generated at a center-of-mass energy of $\sqrt{s} = 14$ TeV using Pythia8 [67], with fast detector simulation by Delphes [68]. The simulation does not incorporate effects from multiple parton interactions or pileup. Jet clustering is carried out using the Anti-$k_T$ algorithm with a radius parameter of $R = 0.8$, based on Delphes E-Flow objects. The dataset includes jets with transverse momentum in the range $p_T \in [550, 650]$ GeV and pseudo-rapidity constrained to $|\eta| < 2$. For top quark events, a valid jet must be located within $\Delta R = 0.8$ of a hadronically decaying top quark, with all three decay products of the top also confined within $\Delta R = 0.8$ from the jet axis. The primary background process considered is QCD dijet production. This dataset consists of one million $t\bar{t}$ events and an equal number of QCD dijet events. Following the standard data split, we allocate 1.2 million events for training, 400,000 for validation, and 400,000 for testing. The dataset is widely used in previous studies, allowing for direct performance comparisons of existing models. [2]

---

[2]It should be noted that there is an effective class imbalance near the top quark mass. The $t\bar{t}$ sample exhibits a pronounced peak around 170 GeV, whereas the mass of the QCD jets is concentrated near zero, resulting in a tiny overlap between their distributions. This disparity makes it particularly challenging to

Table 1: Performance of different networks for top jet classification. Results for networks without an asterisk are taken from their respective references. Plain Transformer was trained from scratch using the structure described in Appendix A.

| | Accuracy | AUC | $1/\epsilon_B(\epsilon_s = 0.5)$ | $1/\epsilon_B(\epsilon_s = 0.3)$ | Parameters |
|---|---|---|---|---|---|
| ParticleNet[54] | 0.940 | 0.9858 | $397 \pm 7$ | $1615 \pm 93$ | 370K |
| PFN[53] | – | 0.9819 | $247 \pm 3$ | $888 \pm 17$ | **86.1K** |
| rPCN[69] | – | 0.984 | $364 \pm 9$ | $1642 \pm 93$ | – |
| **Lorentz invariance based networks** | | | | | |
| PELICAN[35] | **0.9426** | 0.987 | – | $2250 \pm 75$ | 208K |
| LorentzNet[70] | 0.942 | 0.9868 | $498 \pm 18$ | $2195 \pm 173$ | 224K |
| L-GATr[71] | 0.942 | 0.9870 | $540 \pm 20$ | $2240 \pm 70$ | 1.8M |
| **Attention based networks** | | | | | |
| PCT[72] | 0.940 | 0.9855 | $392 \pm 7$ | $1533 \pm 101$ | 193.3K |
| ParT[49] | 0.940 | 0.9858 | $413 \pm 6$ | $1602 \pm 81$ | 2.14M |
| MIParT[50] | 0.942 | 0.9868 | $505 \pm 8$ | $2010 \pm 97$ | 720.9K |
| Mixer[21] | 0.940 | 0.9859 | $416 \pm 5$ | – | 86.03K |
| OmniLearn[73] | 0.942 | **0.9872** | **$568 \pm 9$** | **$2647 \pm 192$** | 1.6M |
| `Plain Transformer`* | 0.938 | 0.9837 | $381 \pm 7$ | $1350 \pm 70$ | 2.1M |
| `IAFormer`* | 0.942 | 0.987 | $510 \pm 6$ | $2012 \pm 30$ | 211K |

Table 1 presents the classification performance of `IAFormer` for top jet classification. `IAFormer` is trained for 23 epochs, with a batch size of 256, with early stopping set for 5 epochs. The AdamW optimizer [74] with an initial learning rate of $5 \times 10^{-4}$ and the Cosine annealing scheduler is used to adjust the learning rate during the training. Two input datasets are considered, particle kinematics, with dimensions $X = (100, 11)$, with a maximum of one hundred particles and 11 features per particle. A second dataset of the pairwise interaction $\mathcal{I}$, which encodes 6 features for each particle pair and total dimension $(100, 100, 6)$. Other networks are trained with the same hyperparameter setup, but trained for 20 epochs. After training, the network performance is evaluated using various metrics, including classification accuracy, Area Under the ROC Curve (AUC), and background rejection at signal efficiencies of 0.3 and 0.5.

`IAFormer` offers performance comparable to other attention-based Transformer networks, while requiring an order of magnitude fewer parameters (211K) than ParT. This reduction in network size is achieved by replacing the $Q$ and $K$ matrices with an interaction matrix, where attention scores are primarily based on pairwise particle interactions. Furthermore, the use of sparse attention enables the network to suppress attention scores of less relevant tokens, reducing the need for excessive model complexity, while efficiently distinguishing between top and QCD jets.

### 3.2.1 The role of sparse attention

The core component of the dynamic sparse attention is the inclusion of the learnable parameter $\beta$, which regulates the level of suppression of less relevant tokens. The left plot in Figure 2 illustrates the distribution of $\beta$ across all `IAFormer` layers. To better understand

---

discern fine-grained differences among high-performance neural networks. Therefore, we use several measures to compare the network performance.

the role of $\beta$, we analyze three different random seeds. Interestingly, all distributions exhibit a similar pattern, $\beta$ values increase in the initial layers, reach a maximum value, and then decrease in the later layers. Moreover, the network classification accuracy improves for higher $\beta$; the blue line corresponds to the best classification accuracy of AUC=0.98678, with $\beta$ value very close to 0.6. Also, it starts at a lower value for earlier layers. This shows that the earlier layer tries to build collective quantities stable against fluctuations, and the rest of the information is abandoned in the later layers dynamically for successful training.

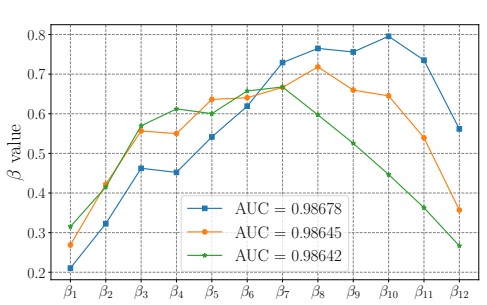 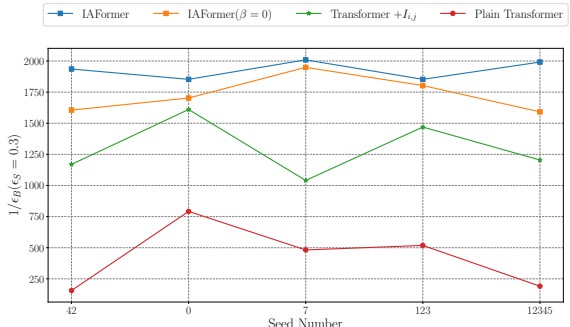

Figure 2: The left plot shows the $\beta$ distribution across the twelve `IAFormer` layers for different seed numbers on the test dataset. The right plot illustrates the uncertainty in the network output when trained with different seed numbers, 42, 0, 7, 123, and 12345. The results of the architectures of the Transformer $+\mathcal{I}_{ij}$(similar to ParT) and the plain Transformer are shown. The uncertainty is reported in terms of background rejection at 0.3 signal efficiency.

The use of sparse attention allows for a substantial decrease in the number of trainable parameters while maintaining the classification performance. This parameter reduction effectively constrains the optimization landscape, leading to smoother convergence dynamics and more stable training behavior. As a result, the model becomes less sensitive to random initialization and stochastic mini-batch fluctuations, which in turn manifests as reduced variance across independent training runs. The right plot of figure 2 illustrates the fluctuation of the background rejection at 0.3 signal efficiency with different seeds for the network parameter initialization. `IAFormer` exhibits robustness against the change of the seed number with a fluctuation range of 150 while plain Transformer has a fluctuation range of 600, and Transformer with interaction matrix (ParT) has a fluctuation range of 550. To further illustrate the impact of sparse attention, we calculate the background rejection for `IAFormer` at $\beta = 0$, which exhibits a fluctuation range of 350. In fact, `IAFormer` resistance to fluctuations arises from its low number of trainable parameters, the structure which achieves higher classification performance through dynamic sparse attention. In contrast, reducing the number of trainable parameters in the other networks typically results in a significant drop in classification performance.

To further evaluate network efficiency, we compute the number of Floating Point Operations (FLOPs) for network layers. This requires understanding the computational complexity of the core operations in both plain Transformer and `IAFormer` models. The primary contributors to the computational cost are the self-attention mechanism and the feed-forward layers. The self-attention operation in a Transformer has a complexity of $O(L^2 \cdot d_{\mathrm{model}})$, where L is the sequence length and $d_{\mathrm{model}}$ is the model dimension. The number of FLOPs, for non-embedding parameters, is defined as $C_{\mathrm{forwrad}} = 2N + 2n_{\mathrm{attn}}d_{\mathrm{model}}n_{ctx}$, with $N$ is

the number of non-embedding parameters in the model, $n_{\text{attn}}$ is the number of the attention layer, $d_{\text{model}}$ is the hidden dimension of the model and $n_{ctx}$ is the number of tokens. As a result, for a particle tokens length, the number of FLOPs for `IAFormer` is 38 million while for the plain transformer is 300 million. These results highlight the substantial reduction in computational cost achieved by `IAFormer`. In particular, the model offers an order-of-magnitude improvement in efficiency compared to the plain Transformer. In addition, the training time per batch of 256 events on an NVIDIA RTX A6000 is 11 seconds for $\beta = 0$ and increases to 16 seconds for $\beta \neq 0$. Although the inclusion of $\beta$ leads to a longer training time, it results in superior classification performance compared to the $\beta = 0$ case.

## 3.3   Quark gluon tagging

The quark jet and gluon jet are expected to show distinguishable features due to different fragmentation patterns arising from the color difference. The quark-gluon tagging is phenomenologically important because BSM particles tend to emit high-energy quarks rather than gluons. The public dataset [66] is generated using Pythia8 [67], where quark(gluon) jets originate from $Z^0$ boson plus quark(gluon) process. The dataset contains not only the four-momentum information but also the particle ID (PID) of jet constituents, which significantly enhances jet identification performance. The PID information is incorporated in a way that aligns with the experimental setup, namely, categorizing particles into five broad types: electrons, muons, charged hadrons, neutral hadrons, and photons, along with their electric charge. These six PID attributes are incorporated as one-hot vectors, and they are combined with the seven kinematic variables, resulting in a total dimension of 13 input features per particle.

The final state particles are clustered into jets using the anti-$k_T$ algorithm with a radius parameter of $R = 0.4$. Only jets with transverse momentum $p_T$ in the range [500, 550] GeV and rapidity $|y| < 2$ are included. The dataset consists of 2 million jets, evenly split between signal and background. Following the recommended partitioning, we allocate 1.6 million jets for training, 200,000 for validation, and 200,000 for testing for `IAFormer` and plain Transformer.

Table 2 presents the performance of `IAFormer` on the quark-gluon tagging task, compared to current state-of-the-art networks. Results for models without an asterisk are taken directly from their published works, while the plain Transformer results are obtained by training the model from scratch. It is important to note that the ParT and MIParT models outperform most other networks, primarily because they apply additional scaling factors to charged and neutral hadrons. Specifically, charged hadrons are weighted as $N(|\text{PID}| == 211) + N(|\text{PID}| == 2212) \times 0.5 + N(|\text{PID}| == 321) \times 0.2$, while neutral hadrons are weighted as $N(|\text{PID} == 130|) + N(|\text{PID} == 2112|) \times 0.5$, where 130, 211, 321, 2112, 2211 stands for $K_L$, $\pi^+$, $K^+$, $n$ and $p$ respectively. These heuristic scaling factors enhance classification accuracy by approximately 1% over the baseline.

For quark-gluon tagging, the number of layers of the `IAFormer` architecture is reduced to six with a total parameter count of 171k. In this scenario, the network performance saturates after a certain number of layers, and adding more layers only introduces redundancy without improving accuracy. The reduction in network size, compared to the top tagging case, may be understood as the result of the similarity between quark and gluon jets. Our choice of the number of layers is based on the distribution of $\beta$ parameters for the attention layers. For `IAFormer` with 12 layers, $\beta$ distribution randomly fluctuates across layers with little improvement in the classification accuracy. By reducing the network size by half, the $\beta$ parameter shows a similar distribution as the top tagging case, as shown in Figure 3. We did not optimize the network parameter further.

Table 2: Performance of different networks on the quark-gluon classification task. Results for networks without an asterisk are taken from their respective references. The plain Transformer was trained from scratch using the architecture described in Appendix A. The results for ParT$_{\text{full}}$ and MIParT$_{\text{full}}$ are shown in italic letters, because they include additional scaling factors for charged and neutral hadrons[49], which are not used in the validation of the other networks.

| | Accuracy | AUC | $1/\epsilon_B(\epsilon_s = 0.5)$ | $1/\epsilon_B(\epsilon_s = 0.3)$ | Parameters |
|---|---|---|---|---|---|
| ParticleNet[54] | 0.840 | 0.9116 | $39.8 \pm 0.2$ | $98.6 \pm 1.3$ | 370K |
| PFN[53] | – | 0.9052 | $37.4 \pm 0.7$ | – | **86.1K** |
| rPCN[69] | – | 0.9081 | $38.6 \pm 0.5$ | – | – |
| **Lorentz invariance based networks** | | | | | |
| PELICAN[35] | **0.8551** | **0.9252** | **$52.3 \pm 0.3$** | **$149.8 \pm 2.4$** | 208K |
| LorentzNet[70] | 0.844 | 0.9156 | $42.4 \pm 0.4$ | $110.2 \pm 1.3$ | 224K |
| **Attention based networks** | | | | | |
| ABCNet[75] | 0.840 | 0.9126 | $42.6 \pm 0.4$ | $118.4 \pm 1.5$ | 230K |
| PCT[72] | 0.841 | 0.9140 | $43.2 \pm 0.7$ | $118 \pm 2.2$ | 193.3K |
| ParT$_{\text{exp}}$[49] | 0.840 | 0.9121 | $41.3 \pm 0.3$ | $101.2 \pm 1.1$ | 2.14M |
| ParT$_{\text{full}}$[49] | *0.849* | *0.9203* | *$47.9 \pm 0.5$* | *$129.5 \pm 0.9$* | 2.14M |
| MIParT$_{\text{full}}$[50] | *0.851* | *0.9215* | *$49.3 \pm 0.4$* | *$133.9 \pm 1.4$* | 720.9K |
| OmniLearn[73] | 0.844 | 0.9159 | $43.7 \pm 0.3$ | $107.7 \pm 1.5$ | 1.6M |
| Plain Transformer* | 0.839 | 0.910 | $39.4 \pm 0.7$ | $109. \pm 1.9$ | 2.1M |
| IAFormer* | 0.844 | 0.917 | $42 \pm 0.3$ | $101 \pm 1.1$ | 171K |

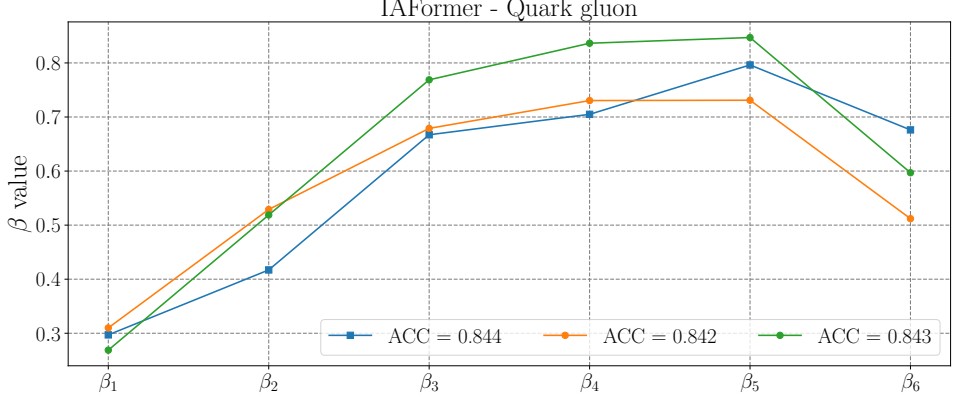

Figure 3: $\beta$ distribution for quark-gluon test data set for three different seed numbers.

Finally, the obtained $\beta$ value of the last layer is larger than the top vs QCD classification of the previous subsections. Because $\beta$ appears negatively in the attentions, the larger $\beta$ likely indicates that the effective degree of freedom required for the quark-gluon classification is smaller than the top-QCD classification.

## 3.4 JetClass Dataset

The JETCLASS dataset [49] is one of the largest and most comprehensive publicly available datasets designed for machine learning applications in jet physics. It was developed to pro-

vide a large scale, standardized benchmark for training and comparing modern deep learning architectures in high energy physics. The dataset is based on simulated proton-proton collision events at a center of mass energy of 14 TeV, generated with PYTHIA8 for event generation and processed through DELPHES3 for fast detector simulation. This ensures a realistic description of both the parton shower and detector response, while maintaining computational efficiency suitable for large scale studies.

The dataset comprises a total of 100 million jets divided equally among ten distinct jet classes, each corresponding to a specific partonic origin or heavy resonance decay topology. These include light quark and gluon jets, heavy-flavor jets, as well as jets originating from hadronic decays of the $W$ and $Z$ bosons, the top quark and the Higgs boson decaying to bottom quarks. Each jet is represented as a collection of its reconstructed constituents, described by low-level particle features such as transverse momentum $p_T$, pseudorapidity $\eta$, azimuthal angle $\phi$, and energy $E$. In addition, high-level jet observables such as jet mass and substructure variables are also provided, allowing for studies that combine low-level and high-level representations within a unified framework.

Table 3: Performance of different networks on the JetClass dataset using only **10M** events.

| | $h\to\bar{b}b$ | $h\to\bar{c}c$ | $h\to gg$ | $h\to 4q$ | $h\to l\nu\bar{q}q$ | $t\to b\bar{q}q$ | $t\to bl\nu$ | $W\to\bar{q}q'$ | $Z\to\bar{q}q$ |
| | $Rej_{50\%}$ | $Rej_{50\%}$ | $Rej_{50\%}$ | $Rej_{50\%}$ | $Rej_{99\%}$ | $Rej_{50\%}$ | $Rej_{99.5\%}$ | $Rej_{50\%}$ | $Rej_{50\%}$ |
|---|---|---|---|---|---|---|---|---|---|
| ParT(10 M) [49] | 8734 | 3040 | 110 | 1274 | 3257 | 12579 | 8969 | 431 | 324 |
| MIParT-L(10 M) [50] | 8000 | 3003 | 112 | 1281 | 3650 | 16529 | 9852 | 440 | 336 |
| IAFormer-L(10 M) | 8264 | 2968 | 114 | 1280 | 3425 | 14314 | 8731 | 423 | 345 |

A key feature of JetClass lies in its unprecedented size and diversity. With ten million examples per category, the dataset provides sufficient statistical power to train deep architectures with millions of parameters. Its inclusion of multiple jet categories ensures that models trained on it can capture a wide range of radiation patterns, color flow structures and multi-prong topologies arising from different SM processes at once. Thus, the task here differs slightly from the Top and quark–gluon datasets, which are trained to distinguish between two distinct classes. In this case, the model must classify among ten different categories simultaneously. Accordingly, we modified the output layer to contain ten neurons.

To evaluate the performance of IAFormer on the JetClass dataset, we train the network on 10 million events and compare its background rejection efficiency with that of other Transformer-based architectures. Motivated by the Chinchilla scaling law for large language models [76], we increase the size of IAFormer to 890k parameters to ensure that the model can effectively capture the rich information contained in the JetClass dataset. Table 3 summarizes the background rejection efficiencies for all classes, comparing IAFormer-L with MIPart-L and ParT, when trained on a dataset of size 10M.

# 4    Analysis of hidden layer representations

We now compare the attention-based transformer models (Plain Transformer, Transformer + $\mathcal{I}_{ij}$(ParT), and IAFormer) using several interpretability methods to study how hidden layers are organized for classification. Different interpretability methods can be applied to analyze attention-based Transformer models designed for particle cloud data. These techniques provide insights into how the model processes and prioritizes different parti-

cles, helping to understand learned representations and the network decision-making. By leveraging these interpretability techniques, one can gain deeper insights into the behavior of attention-based Transformers, ensuring that their predictions align with meaningful physical patterns. Common interpretation methods are

- **Attention Maps:** These maps visualize how the Transformer model distributes its attention across different particles in the cloud [77]. By highlighting the most influential particles, attention maps provide an intuitive understanding of the pair of tokens relevant to the classifications.

- **CKA Similarity:** Centered Kernel Alignment (CKA) measures the similarity between hidden layer representations within a model or across different models [78]. A high CKA similarity between layers indicates redundancy, suggesting that certain layers can be pruned without significant loss of performance, while a low CKA similarity suggests that layers learn distinct features that contribute to improved classification accuracy.

- **Saliency Maps:** These maps measure the sensitivity of the model's output to small perturbations in the input [79] model saliency maps help assess whether the model captures physically meaningful features or relies on spurious correlations.

- **Layer-wise Relevance Propagation (LRP):** LRP assigns relevance scores to individual particles, quantifying their contribution to the final decision [80]. This technique helps trace back model predictions to specific input particles. Different LRP variants can be used to enhance interpretability for high-energy physics applications.

- **Grad-CAM:** This method generates class-specific activation maps by computing the gradients of the predicted class score with respect to the final Transformer layer [81]. For example, Grad-CAM can highlight crucial regions in the $eta - \phi$ plane, allowing for a geometric interpretation of how the model distinguishes between different classes.

In the following, we focus on attention maps and CKA similarity to interpret the results obtained from the top tagging. The attention maps are utilized to visualize the sparse pattern in the attention score when using `IAFormer`, while the CKA is used to explore the learned representation by the different attention layers.

## 4.1   Attention Maps

The attention map is a 2D heatmap of attention scores that highlights the model focus. To compute attention maps, the attention scores, which represent the strength of the relationship between different elements of the input, are computed by taking the dot product between the query vector of one element and the key vector of every other element in the input. These scores are then normalized using a softmax function, which ensures that they sum to one. The final attention map is a visual representation of these scores, showing which parts of the input the model "attends to" or focuses on most when producing its output. In `IAFormer`, the attention map is computed as the difference between the learnable pairwise interaction matrix in equation 9.

Figure 4 presents the attention maps from the final layer of `IAFormer` and a plain Transformer, using 10,000 test events. In both networks, the final layer consists of 16 attention heads, displayed individually. In these maps, the $X$ and $Y$ axes represent the particle tokens, hadrons within the jet, while the color scale indicates the attention score

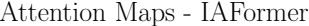

Attention Maps - IAFormer          Attention Maps - Plain Transformer

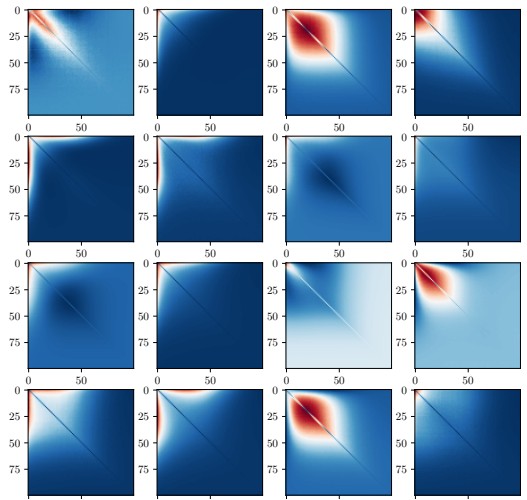 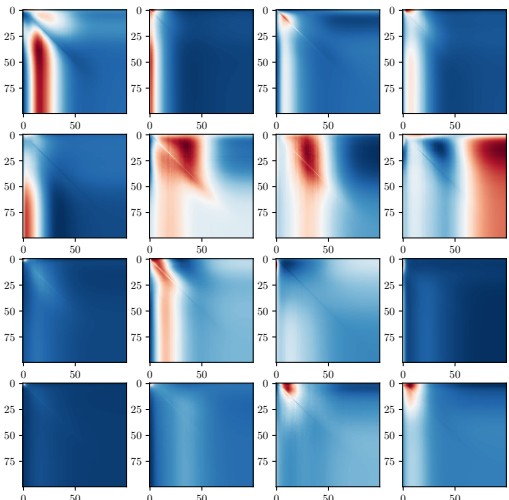

Figure 4: Attention maps of the final self-attention layer of `IAFormer` (left) and a plain Transformer (right), generated using 10,000 test events from the top jet dataset. Each network comprises 16 attention heads, shown individually. The axes represent particle tokens, with each event containing 100 particles, while the color bar denotes the attention scores assigned to each particle pair, ranging from 0 to 1.

assigned to each particle pair. It should be noted that the tokens do not exactly correspond to the particle inputs, because the token of the final layer is a mixture of input particles.

Compared with a plain Transformer model, the attention maps of `IAFormer` show high attention scores concentrated on a smaller number of final tokens across all attention heads, with the remaining particles significantly suppressed. This shows `IAFormer`s more efficiently accumulate relevant information from earlier tokens. Ideally, the network should focus on the cluster of hadrons that form the three-prong structures of the top jet. If the network assigns high attention to a large number of tokens, it can hinder classification performance. Additionally, `IAFormer` attention maps exhibit a symmetric pattern over the diagnoal, reflecting the symmetric nature of the pairwise interaction matrix. This symmetry arises even though we do not require the symmetry to weight factors $W_1$ and $W_2$.

## 4.2 CKA similarity

One of the key challenges in studying neural network representations is that features are distributed across multiple neurons, and the number of neurons in hidden layers often exceeds the input dimension while varying across layers and models. This variability makes direct comparisons difficult. Centered Kernel Alignment (CKA) similarity, derived from kernel methods and alignment-based metrics, provides a powerful tool for assessing the independence of the captured information across the different layers. Traditional similarity measures, such as Pearson correlation or Euclidean distance, directly compare the raw hidden sector values. Instead of comparing the raw values, CKA evaluates how well learned "representations" in the hidden sector align in high-dimensional feature space, and therefore, it is more robust. The CKA similarity can capture intricate, nonlinear relationships between features, making it particularly effective for analyzing deep learning models.

To compute CKA similarity, consider two activation matrices, $X \in \mathbb{R}^{d \times P_1}$ and $Y \in \mathbb{R}^{d \times P_2}$, where $d$ represents the number of input samples, and $P_1$ and $P_2$ denote the number of neurons in two different hidden layers. The CKA similarity is defined as

$$\mathrm{CKA}(M, N) = \frac{\mathrm{HSIC}(M, N)}{\sqrt{\mathrm{HSIC}(M, M)\mathrm{HSIC}(N, N)}}, \tag{12}$$

with $M = XX^T$ and $N = YY^T$ are the Gram matrices of the two hidden layers, both of size $d \times d$. Since the Gram matrix dimensions depend only on the number of input samples, CKA allows for the comparison of hidden layers with different neuron counts, as well as representations from different models. The Hilbert-Schmidt Independence Criterion (HSIC) measures statistical dependence between two matrices and is defined as

$$\mathrm{HSIC}(M, N) = \frac{1}{(d-1)^2}\mathrm{Tr}(MHNH), \tag{13}$$

where $H_{ij} = \delta_{ij} - \frac{1}{d}$ is a centering matrix, which ensures that each row and column sums to zero for $A = M, N$. Centering prevents CKA from being biased by extreme values or outliers, ensuring more stable and meaningful comparisons.

CKA values range from 0 to 1, where higher values indicate strong similarity between learned representations. If two consecutive layers have high CKA similarity, it suggests that the second layer does not significantly enhance classification accuracy, implying that it could be removed without affecting performance. On the other hand, layers with low CKA similarity capture distinct aspects of the data, contributing to improved model performance by learning complementary features.

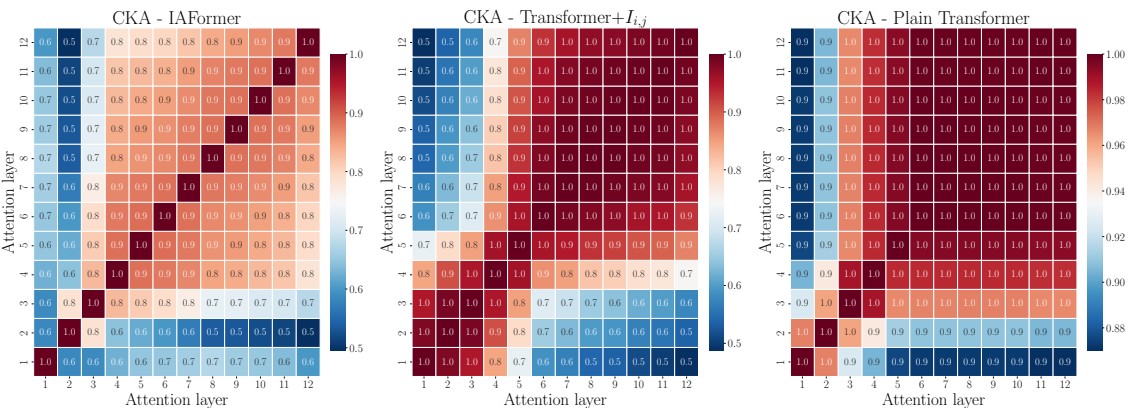

Figure 5: Linear CKA similarity for `IAFormer` (left), Transformer $+\mathcal{I}_{i,j}$ (middle), and Plain Transformer (right) using 1000 test events from the top jet dataset. The axes represent the attention layers in each network, while the colour bar indicates the CKA values.

We compute the output from each attention layer with dimensions $(100, 32)$, where the first dimension corresponds to the number of particles and the second represents feature embeddings. To construct the Gram matrices, $M$ and $N$, we consider 1000 test events and average over the feature dimension, resulting in Gram matrices of size $(1000, 1000)$. These matrices are then used to compute the CKA values for three different networks, `IAFormer`, Transformer with an interaction matrix( similar to ParT), and Plain Transformer, as shown in Figure 5. For a fair comparison, all networks are trained with 12 attention layers, represented on the $X$ and $Y$ axes of each plot.

Attention layers of `IAFormer` exhibit lower CKA values, particularly in the first four layers, with CKA values ranging from 0.5 to 0.8. This suggests that these layers capture different patterns in the jet constituents, contributing to an overall improvement in classification performance. The later layers exhibit stronger internal similarity, with CKA values exceeding 0.8. The CKA structure indicates an efficient flow of attention from the early layers to the final ones. The middle plot presents the CKA values for the Transformer incorporating the interaction matrix. The CKA matrix exhibits a block-diagonal structure, where the first four layers demonstrate strong internal similarity but differ significantly from the remaining layers. Additionally, the last four layers encode nearly identical information, with a CKA value of 1. In contrast, Plain Transformer shows consistently high similarity across all layers, with CKA values ranging from 0.85 to 1. This uniformity likely contributes to its lower performance in the top tagging task compared to other networks.

Overall, Plain Transformer and Transformer+ $\mathcal{I}_{ij}$ stop improving the first 6 layers, while `IAFormer` steadily improves for all layers, with stronger suppression of the fluctuation by larger $\beta$ layer by layer, likely building global variables along the increasing layers.

## 5 Conclusion

In this paper, we introduce `IAFormer`, an attention-based deep learning network that incorporates dynamic sparse attention to reduce network size while preserving classification performance. Furthermore, the standard attention weights are replaced with pairwise interactions between input particles. This ensures the attention matrix preserves the invariance under boost along the beam direction. Combined with sparse attention, this modification reduces the redundancy typically present in standard Transformer architectures.

To evaluate the network performance, we validate `IAFormer` on two benchmark tasks, top quark tagging and quark–gluon discrimination. In both cases, `IAFormer` achieves improved classification performance compared to ParT network, while significantly reducing model size. Furthermore, `IAFormer` delivers performance that is competitive with state of the art Lorentz-invariant networks.

To further analyze the results, we employ various AI interpretability techniques to visualize the information learned by `IAFormer` hidden layers in comparison to other Transformer-based architectures. Specifically, we utilize attention maps and CKA similarity. The attention maps reveal `IAFormer` condenses the information efficiently in fewer tokens, in contrast to the scattered attention patterns observed in the original Transformer, where weights are more uniformly distributed across all particle tokens. This shows that the dynamic sparse attention enables `IAFormer` to concentrate on the most relevant tokens and suppress attention to less informative inputs. The behavior is not exhibited by the original Transformer, which tends to assign attention broadly, even to less important tokens. Additionally, CKA analysis indicates that the similarity between the learned representations among different layers of `IAFormer` is lower than that of other Transformer variants, highlighting its unique learning dynamics.

Although `IAFormer` demonstrates improved performance over existing Transformer-based networks, its architecture still requires optimization for specific tasks. For instance, the learnable parameter $\beta$ can exceed 1, which amplifies noise in the attention weights. In `IAFormer`, the value of $\beta$ is clipped to a maximum of 1 to mitigate this effect. Additionally, it is possible for attention scores to become negative, making them unsuitable for interpretation as attention probabilities.

In fact, it may be worthwhile to consider the physical interpretation of the trainable parameter $\beta_i$. These hyperparameters depend on the underlying probability distributions of

the signal and background, and can be viewed as global variables that quantify the distinction between different jet classes. In particular, the values of $\beta$ in the final layers are likely associated with the effective degrees of freedom necessary to capture the differences between signal and background, assuming the network depth has been appropriately optimized.

To better understand the role of the trainable suppression parameter $\beta$ in `IAFormer`, we added a regularization term to the loss that discourages sharp changes in $\beta$ across layers. This helps the model learn a smoother, more structured evolution of $\beta$, guided by layer-wise scaling factors. We found that this leads to the same layer-wise $\beta$ pattern reported earlier, a rise and fall across layers, which supports its interpretation as reflecting the effective degrees of freedom needed to separate signal from background. This reinforces the idea that $\beta$ captures physically meaningful information that shapes the model's sparse attention behavior.

Finally, various `IAFormer` architectures tailored for different classification tasks have been developed and made publicly available.

## Acknowledgments

This work is funded by grant number 22H05113, "Foundation of Machine Learning Physics", Grant in Aid for Transformative Research Areas and 22K03626, Grant-in-Aid for Scientific Research (C). WE is funded by the ErUM-WAVE project 05D2022, "ErUM-Wave: Antizipation 3-dimensionaler Wellenfelder", which is supported by the German Federal Ministry of Education and Research (BMBF).

## A  Code Availability

Although"`IAFormer` is validated against the current DL networks for jet tagging problems, it is adopted for generic classification problems. Additionally, our code comprises three different networks that can be used by public users; `IAFormer`, Plain Transformer, and Transformer with interaction matrix(similar to ParT). In this section, we present the full details on how to use the implemented networks for different classification tasks. Implementation of the `IAFormer` model is publicly available at `IAFormer` (https://github.com/wesmail/IAFormer). The repository is implemented using PyTorch Lightning[3], a high-level deep learning framework built on top of PyTorch [82]. It offers a modular, scalable, and user-friendly interface for training and evaluating deep learning models.

The repository is organized into several key components:

- **utils/graph_builder.py**: A module for transforming raw jet constituent data into an input suitable format for the `IAFormer`.

- **models/**: Contains the implementation of the `IAFormer` model, including the implementation of vanilla multi-head attention and sparse attention.

- **data/**: Implementation of data loading and preprocessing.

- **configs/**: YAML configuration files for different transformer based models.

- **callbacks/**: Implementation of inference and plotting.

---

[3]https://github.com/Lightning-AI/pytorch-lightning

Installation instructions and guidelines for training and evaluating different models are provided in the GitHub repository's `README.md`. Users can easily reproduce results or experiment with custom configurations by modifying the corresponding YAML files. Training the network is done via the following command

```
python main.repository'snfig=configs/ia_former.yaml
```

The code is equipped with three different configuration files for the different networks. Hyperparameters of the best performance are saved in the checkpoints directory. To test the network

```
python main.py test -config=configs/ia_former.yaml -ckpt_path=best_point.ckpt
```

In the following, we describe the individual components and modules that constitute the `IAFormer` implementation.

## A.1 Particle Graph Construction

We have implemented two dedicated preprocessing classes: `TopParticleGraphBuilder` and `QGParticleGraphBuilder` for the two different datasets used in this article, the top tagging dataset and the quark-gluon dataset. Each class performs the same task, but is tailored to the input format of its respective dataset.

Each class reads the four-momentum components (`PX`, `PY`, `PZ`, `E`) of the jet constituents for each event, computes derived kinematic quantities and pairwise interaction features, and stores the output in an HDF5 file, which contains the following datasets:

- `feature_matrix`: A 3D array of shape $[N_{\text{events}}, N_{\text{particles}}, F]$. Here, $F$ are the features, which include the particle four momenta, log-scaled transverse momentum and energy, relative kinematics, spatial distances (Eq. 10), and particle identification variables, for the quark-gluon dataset.

- `adjacancy_matrix`: A 3D array of shape $[N_{\text{events}}, (N_{\text{particles}} \cdot (N_{\text{particles}} - 1))/2, 6]$ that contains pairwise interaction features between particles, which are listed in Eq. 11.

- `mask`: A 2D boolean array of shape $[N_{\text{events}}, N_{\text{particles}}]$ indicating valid (non-padded) particle entries in each event.

- `labels`: A 1D array of shape $[N_{\text{events}}]$ that stores the ground-truth class labels.

## A.2 Data Handling Module

The `data_handling.py` module provides functionality for loading the preprocessed HDF5 files generated by the `graph_builder.py` module and integrating them with PyTorch Lightning workflow. The data loading class **H5Dataset** implements the `reconstruct_adjacency()` method, which infers the number of particles from the number of pairwise entries and reconstructs the dense adjacency matrix accordingly.

## A.3 Networks structure

The core classification model used in `IAFormer` is implemented in the `Transformer`. It supports multiple attention mechanisms mentioned above via the `attn` argument. The training logic is handled by the `JetTaggingModule`, a PyTorch Lightning module that encapsulates the training, validation, and testing procedures.

**Transformer Class**

The `Transformer` model consists of the following components:

- **Input Embeddings**:
  - `ParticleEmbedding`: Projects node-level features into a common embedding space using an MLP.
  - `InteractionInputEncoding`: Projects edge-level features ($\mathcal{I}_{i,j}$) into a common embedding space using a series of `Conv2D` layers.

- **Attention Blocks**: A stack of `ParticleAttentionBlock` modules.

- **Feature Aggregation and Output**: The per-particle representations are pooled across the feature dimension and passed to a linear classification head for binary classification.

**JetTaggingModule**

The `JetTaggingModule` is a PyTorch Lightning wrapper around the `Transformer` model. It defines the full training and evaluation pipeline:

- `training_step()`: Computes binary cross-entropy loss on the logits and logs training accuracy.

- `validation_step()` and `test_step()`: Evaluate the model on validation and test datasets, respectively, computing accuracy, loss, and the area under the ROC curve.

During testing, attention weights and intermediate activations from each transformer block are collected for later downstream analysis. The following arguments control the architecture and training process:

- `embed_dim`: Dimension of the embedding space for particles.

- `num_heads`: Number of attention heads per block.

- `num_blocks`: Depth of the transformer.

- `max_num_particles`: Maximum number of particles per event.

- `attn`: Type of attention used (`plain`, `interaction`, or `sparse`).

## A.4 Attention Module

The `attention.py` module implements three types of attention mechanisms used in this article:

- **Plain Attention**: A standard scaled dot-product attention without any interaction features.

- **Interaction Attention**: Extends plain attention by incorporating an interaction matrix into the attention logits before the softmax operation.

- **Sparse Differential Attention**: A novel attention mechanism that combines two interaction-aware attention maps using a learnable $\beta$.

**MultiHeadAttention**

The `MultiHeadAttention` class implements standard multi-head attention. It supports an optional interaction matrix $\mathcal{I}_{i,j}$ that is projected using a `Conv2D` layer to match the shape of the attention logits. If provided, this interaction matrix is added to the dot-product attention scores before applying softmax. An optional attention mask (`umask`) can be used to ignore invalid or padded elements. The attention mask offers the advantage of excluding padded particle tokens from the attention computation. In practice, if an event contains fewer particles than the maximum sequence length, the remaining positions are padded with zeros. Without masking, these zero-padded tokens may receive nonzero attention scores, e.g. 1, after the softmax. To address this, an attention mask is applied to assign a value of $-\infty$ to the padded tokens before the softmax operation. As a result, their attention scores become effectively zero after the softmax function, ensuring they do not influence the output.

The module consists of:

- Linear projections for queries, keys, and values: `q_proj`, `k_proj`, `v_proj`.

- A learned projection for the interaction matrix: `u_proj`.

- A final output projection: `out_proj`.

**MultiHeadSparseAttention**

The `MultiHeadSparseAttention` class implements a differential attention mechanism. The attention map is calculated entirely from the projected interaction matrix $\mathcal{I}_{i,j}$ using a `Conv2D` layer. Two attention maps are extracted and combined via a learnable scalar $\beta$, parameterized through dot products between $\beta_q$ and $\beta_k$. This allows the model to softly subtract one attention map from another, enabling a sparse or selective focus on particle-particle interactions.

- The learnable $\beta$ coefficient is initialized using a depth-dependent function: `beta_init`.

- The attention weights are computed as:

$$\text{attn} = \text{Softmax}(\mathcal{I}_{i,j})[:,:,0] - \beta \cdot \text{Softmax}(\mathcal{I}_{i,j})[:,:,1]$$

**ParticleAttentionBlock**

The `ParticleAttentionBlock` class is a modular block that wraps the attention mechanism with normalization and a feed-forward MLP:

- It supports all three attention variants via the `attn` argument (`plain`, `interaction`, `diff`).

- Applies residual connections and normalization both before and after the attention sub-layer.

- Includes an MLP with GELU activation and expansion factor.

## A.5   JetClass IAFormer Implementation

This subsection describes the specific implementation of `IAFormer` for the JetClass benchmark, including data conversion, storage format, data loading, and model training.

**JetClass Data Conversion and Storage**

The JetClass dataset is provided in ROOT format, containing per-jet constituent information with variable-length particle lists. To enable efficient large-scale training, we implement a dedicated ROOT-to-HDF5 conversion pipeline. The converter reads jet constituent four-momentum components and auxiliary features, constructs particle-level node features and pairwise interaction features, and stores the result in sharded HDF5 files.

Each HDF5 file contains the following datasets:

- `node_features`: A tensor of shape $[N_{\text{events}}, N_{\text{particles}}, F]$, storing per-particle features such as kinematic variables and derived quantities.

- `adjacency_matrix`: A compact upper-triangular representation of pairwise interaction features with shape $[N_{\text{events}}, N_{\text{pairs}}, U]$, where $N_{\text{pairs}} = N_{\text{particles}}(N_{\text{particles}} - 1)/2$.

- `labels`: A one-dimensional array of integer class labels for each event.

- `n_particles`: The true number of particles per event before padding.

Storing the interaction matrix in upper-triangular form significantly reduces disk usage and I/O overhead while preserving all pairwise information required by interaction-aware attention. Dataset-level metadata (feature definitions, maximum particle count, class mapping) are stored as HDF5 attributes for reproducibility.

**Data Handling and Graph Reconstruction**

The `H5Dataset` class in `data/` provides efficient access to the JetClass HDF5 files and integrates directly with the PyTorch Lightning `DataModule`. During loading, the dataset:

- infers the number of particles from `n_particles`,

- reconstructs the dense or sparse particle-particle adjacency structure from the stored upper-triangular representation,

- builds `edge_index` and `edge_attr` tensors compatible with PyTorch Geometric-style batching,

- applies masking to padded particles to ensure they do not contribute to attention or pooling.

To support large-scale JetClass training, HDF5 file handles are cached using an LRU strategy, and static graph structures (e.g. upper-triangular indices) are cached once per process.

**Model Architecture for JetClass**

For JetClass classification, `IAFormer` operates on particle sequences with explicit interaction modeling. The model consists of:

- **Particle Embedding**: An MLP that projects node-level particle features into a common embedding space.

- **Interaction Encoding**: A learnable projection of pairwise interaction features, used to modulate attention scores.

- **Transformer Blocks**: A stack of particle-attention blocks supporting plain, interaction-aware, or sparse differential attention.

- **Pooling and Classification**: Masked pooling over particle embeddings followed by a linear classification head for multiclass jet tagging.

The choice of attention mechanism (`plain`, `interaction`, or `diff`) is controlled via the `attn` argument in the configuration files.

**Training and Evaluation**

Training, validation, and testing are implemented in a dedicated PyTorch Lightning module. The module computes the cross-entropy loss for multiclass classification and logs standard performance metrics, including accuracy and area under the ROC curve. During testing, attention maps and intermediate representations can optionally be stored for downstream interpretability studies.

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

Dear Editor,

The authors thank the referee for reading our manuscript and providing valuable comments. We have made the appropriate changes based on the comments of the referee. We have addressed all the points raised by the referee and believe that our manuscript is now ready for publication. With best regards,

The authors

# Response to the first Referee

The referee's comments are reproduced in italic font.

1. " *"First, the number of attention heads has to be increased with the number of features of the interaction matrix." In the original ParT a linear layer + non-linear activation is used to project the interaction matrix to whatever is the desirable dimension needed, so this shortcoming is not true."*

   **Response:** In the standard Transformer layer, the multi-head attention score matrix is expressed as
   $$\text{Attention score} = \text{softmax}\left(\frac{Q^h(K^T)^h}{\sqrt{d}}\right),$$
   where $h$ denotes the number of attention heads. In this formulation, $Q \cdot K^T$ has dimensions (batch size, number of particles, number of particles, $h$). The pairwise interaction matrix, has dimensions (batch size, number of particles, number of particles, number of features) To incorporate this interaction matrix into the attention score as

   $$\text{softmax}\left(\frac{Q^h(K^h)^T}{\sqrt{d}} + \mathcal{I}_{i,j}\right),$$

   the number of attention heads must match the number of features in the pairwise interaction matrix.

   As ParT includes a feature embedding layer that can adjust the feature dimension to align with the number of attention heads, projecting high-dimensional features into a lower-dimensional space may lead to information loss. We modified this sentence to "The feature dimension of the interaction matrix needs to be reshaped to be compatible with the specified number of attention heads."

2. " *"Second, all Transformer layers use the same fixed interaction matrix as a bias, preventing the model from learning updated representations in deeper layers, leading to lower performance." it is true that the bias is the same, but the rest of the attention coeffients are allowed to change in the attention matrix, so it's also not true that the model cannot learn updated representations (if that was true using multiple transformer blocks wouldn't help!!!)."*

   **Response:** By this statement we meant that this type of structure incorporate the same pairwise interaction matrix to all Transformer layers. On the other hand, IAFormer setup, the interaction matrix is updated at each layer and this updated interaction matrix is

passed to the later IAFormer layers. To avoid the confusion, we deleted this sentence from the draft.

3. *" "The IAFormer architecture comprises a data embedding block, attention-based layers, and a final MLP for classification, as can be seen in Figure ." No figure number."*

**Response:** Has been corrected in the text.

4. *" To me it is not clear why the sparse attention is needed, as it should increase the total number of FLOPs in each transformer block by a large margin as you need to apply 2 linear transformations to the interaction matrix. The other important benchmark to be added is simply one where beta is 0, which should help understand the benefit from the sparse implementation."*

**Response:** The motivation for introducing sparse attention in our model is to enhance the stability of the attention mechanism by explicitly suppressing irrelevant or noisy token interactions. While sparse attention introduces additional linear transformations, these operations act on a reduced set of pairwise interactions, as irrelevant tokens are highly suppressed almost approaching zero. Consequently, the overall computational overhead remains moderate compared to a dense attention mechanism applied to all particle pairs. In addition, we were able to reduce the network size without sacrifice the network performance.

To address the referee suggestion, we added a subsection 3.2.1 including an additional benchmark where $\beta = 0$, including a discussion about the FLOPs and training time.

5. *"Similarly, can you add the number of FLOPs for each baseline used? The number of parameters is fine if the main bottleneck is memory, but the number of flops is the quantity that actually tells how feasible is to use these algorithms."*

**Response:** The number of the FLOPs is porportional to the number of the network parameters. According to ArXiv:2001.08361, the number of FLOPs, for non-embedding parameters, is defined as

$$C_{\text{forwrad}} = 2N + 2n_{\text{attn}}d_{\text{model}}n_{ctx}\,, \tag{1}$$

with $N$ is the number of non-embedding parameters in the model, $n_{\text{attn}}$ is the number of the attention layer, $d_{\text{model}}$ is the hidden dimension of the model and $n_{ctx}$ is the number of tokens. From this formula we can see that the size of the network parameters can give a direct indiction about the number of the FLOPs.

As the same comment raised by the two referees, we compare the FLOPs of IAFormer and Plain Transformer in the newly added subsection 3.2.1.

6. *"The top tagging dataset is commonly used but it is a clearly saturated dataset for most applications. Can you show the results for bigger datasets such as JetClass? On the same note, the main benefit from transformers are the scalability with the dataset size. By using a bigger dataset you can show how does your transformer model compare against a standard transformer or ParT."*

**Response:** We added a subsection discussing the IAFormer performance on JetClass dataset with extending the appendix to describe the new implementation. As for our limited allowed computational resources, we compare the `IAFormer` performance with Part and MIPart using **10M** events.

7. *'"IAFormer outperforms other attention-based Transformer networks, although it has an order of magnitude smaller parameter size of 211K than ParT" This is not true. L-GATr, which is also a transformer, shows better performance. OmniLearn, which is also a transformer, the same (although it is pre-trained, so you could make a distinction), but more relevant, IAFormer has the same performance as MIParT, which is very relevant and oddly omitted for these results since it is shown later on in the case of QG separation."*

   **Response:** We have changed this sentence to "`IAFormer` offers performance comparable to other attention-based Transformer networks, while requiring an order of magnitude fewer parameters (211K) than ParT."

8. *"Table 1: Often bold results are used to highlight the best results, not the authors results. That helps the reader to quickly contextualize the new model performance compared to current SOTA. I would use the bold to show the best results for each metric and add "IAFormer (this work)" to the text. Same for Table 2."*

   **Response:** We modified the draft to address this comment.

9. *'"Furthermore, the use of sparse attention enables the network to suppress attention scores of less relevant tokens, reducing the need for excessive model complexity, while efficiently distinguishing between top and QCD jets. " I'm not convinced the reduced variance of the network outputs is necessarily accounted by the sparse attention. For this claim you either need to set beta to 0 to show the case of no sparsity, which is not one of the benchmarks, or possibly implement the sparse attention in the baseline transformer model, to show that in that case the variance of the output is reduced. Moreover, there are additional choices that influence the fluctuations of the output, such as the choice of optimizer, learning rate, convergence criteria, learning rate schedule, so on. While I agree that IAFormer is better than the baselines, I don't think there's enough evidence that the sparse attention improves the stability of the outputs."*

   **Response:** It is important to note that the reduced variance is not caused by the sparsity mechanism alone, but rather by the combined effect of the sparse design making the network more efficient and the resulting regularization on the training process. To address the referee suggestion, we added a subsection 3.2.1 including an additional benchmark where $\beta = 0$.

10. *"Table 2: The results shown are a bit misleading because the final performance highly depends on the choice of parameterization of the PID, as the authors briefly mentioned. If only experimentally accessible PIDs are used (which I assume is the case for the authors) you get 1/bkg eff at 30% sig eff of around 100 while if you also split the PIDs into all possible categories you get the results where this same metric is 130. Either the authors split the models between these 2 categories, or evaluate their model based on both options, otherwise the comparison is not fair since the difference of 1% the authors mentioned is only true for metrics like AUC but if you look at PELICAN, it's better than IAFormer by*

50% in the 30% sig eff metric, which again has nothing to do with the network itself. Additionally, the authors omit other models with benchmarks in this dataset such as ABCNet (https://arxiv.org/abs/2001.05311) and PCT (https://arxiv.org/abs/2102.05073), which are also transformer based models."

**Response:** We appreciate the referee comment. However, we believe that addressing this issue in the text effectively conveys the idea, while training both IAFormer and Plain Transformer on two different cases would be too cumbersome to present in the table. Additionally, we have clearly stated that these performance results cannot be directly compared to those of other networks. Missing references have been added to the result tables (Table 2 and Table 3).

11. *"In Fig. 4 how are the particles ordered? I agree that the diagonal has more structure for IAFormer, but the particle ordering is arbitrary, so I do not understand the symmetry argument used (you could swap rows and columns in the 2D maps resulting in a valid attention map but more messy)."*

**Response:** For all input datasets, particles are ordered with their $P_T$. By symmetry, we meant the symmetric structure of the attention maps over the diagonal which arises from our replacement of the $Q.K^T$ term with a pairwise interaction matrix. Since this matrix is symmetric by design, it naturally results in symmetric attention patterns. The sentence is modified to avoid much clarence.

# Response to the second Referee

The referee's comments are reproduced in italic font.

1. *"The main motivation for using transformers for jet tagging is that they scale well to large datasets while keeping manageable computational cost even for millions of parameters. The authors benchmark IAFormer only on the small-scale top tagging and quark-gluon datasets, where message-passing graph networks and transformers achieve similar performance to transformers. To demonstrate that IAFormer is useful for large-scale datasets, the authors should evaluate IAFormer on the JetClass dataset, and perhaps also a IAFormer-L with 2M parameters (similar to MIParT-L)."*

**Response:** We added a subsection discussing the IAFormer performance on JetClass dataset. As for our limited allowed computational resources, we compare the `IAFormer` performance with Part and MIPart using **10M** events.

2. *"The authors write in the abstract "Despite being computationally efficient by more than an order of magnitude than the Particle Transformer network" and probably mean that IAFormer has 10x less parameters than ParT. While small parameter count makes interpretability easier and avoids overfitting, it has little to do with computational efficiency. For instance, a transformer with 2M parameters typically trains faster and uses less memory than a 200k parameter fully-connected graph network. To complete the picture, the authors should compare FLOPs and memory consumption of their plain transformer and IAFormer, and ideally also compare the timing on the same GPU."*

**Response:** We agree with the referee that different ML layers have different computational efficiency. Although, when comparing transformer based layers together the computational efficiency mainly proportional to the number of trainable parameters, as per our reply on comment 5 by the first referee. In the modified version of the draft we compare the FLOPs and training time between IAFormer and plain Transformer in subsection 3.2.1.

3. *"IAFormer combines two ingredients, (a) differential attention and (b) using only edge features in attention. The two features do not rely on each other, raising the question of which aspect gives the performance gain. The authors already include a plain transformer baseline, making it natural to also include the seperate cases of only (a) and only (b) in addition to the full IAFormer (a)+(b) for one dataset, e.g. top-tagging. To increase the value of this ablation, the authors should optimize the training hyperparameters and regularization of the plain Transformer, they can e.g. use the ParT training hyperparameters. With a good training, the plain Transformer should achieve AUC values in the range 0.9840 - 0.9850. This study would be very valuable for the community, as differential attention is be easy to implement also in other transformer architectures in HEP. But this is less relevant than the two points above."*

**Response:** We have optimized the plain Transformer which achieves higher classification performance in the modified draft.

4. Minor points:

   (a) Figure 3 corrected.
   (b) We thank the referee for his suggestion. We have included sigmoid function instead of $\beta$ clipping.
   (c) We report all measures to three digits places to remain consistent with the published work. Also, we have added the number of parameters for L-GATr to Table 1.
   (d) Text has been corrected.
   (e) The use of class tokens is further clarified in the footnote on page 8. Since ParT relies on class tokens, we have kept this sentence.
   (f) Results for $Part_{exp}$ is added to Table.2 .
   (g) Naming convention has been unified.