# Peer review of "IAFormer: Interaction-Aware Transformer network for collider data analysis"

_SciPost Physics_

## Round 1 · Referee Report · Anonymous (Referee 1) · 2025-8-13

Report

The authors introduce a new model architecture named IAFormer. The new model builds on top of previously successful models for particle tagging, where the use of interaction matrices can greatly improve tagging performance. In particular, the authors identify an alternative approach to encode the interaction matrix resulting in their proposed architecture. The paper is well written and clear. While results in widely used benchmarks were presented, is has been widely noticed by the HEP community that older benchmarks are limited by the dataset sizes and complexity. Nowadays there are much bigger datasets available, such as the ATLAS top tagging dataset and the JetClass dataset. Since transformers are revered for their scaling properties, it's a must to evaluate their models on these datasets. I'm in favor of the publication of this article if the additional benchmarks are used and the detailed comments below are addressed.

Requested changes

1: “First, the number of attention heads has to be increased with the number of features of the interaction matrix.” In the original ParT a linear layer + non-linear activation is used to project the interaction matrix to whatever is the desirable dimension needed, so this shortcoming is not true.

2: “Second, all Transformer layers use the same fixed interaction matrix as a bias, preventing the model from learning updated representations in deeper layers, leading to lower performance.” it is true that the bias is the same, but the rest of the attention coeffients are allowed to change in the attention matrix, so it’s also not true that the model cannot learn updated representations (if that was true using multiple transformer blocks wouldn’t help!!!)

3: “The IAFormer architecture comprises a data embedding block, attention-based layers, and a final MLP for classification, as can be seen in Figure .” No figure number

4: To me it is not clear why the sparse attention is needed, as it should increase the total number of FLOPs in each transformer block by a large margin as you need to apply 2 linear transformations to the interaction matrix. The other important benchmark to be added is simply one where beta is 0, which should help understand the benefit from the sparse implementation

5: Similarly, can you add the number of FLOPs for each baseline used? The number of parameters is fine if the main bottleneck is memory, but the number of flops is the quantity that actually tells how feasible is to use these algorithms.

6: The top tagging dataset is commonly used but it is a clearly saturated dataset for most applications. Can you show the results for bigger datasets such as JetClass? On the same note, the main benefit from transformers are the scalability with the dataset size. By using a bigger dataset you can show how does your transformer model compare against a standard transformer or ParT.

7: “IAFormer outperforms other attention-based Transformer networks, although it has an order of magnitude smaller parameter size of 211K than ParT” This is not true. L-GATr, which is also a transformer, shows better performance. OmniLearn, which is also a transformer, the same (although it is pre-trained, so you could make a distinction), but more relevant, IAFormer has the same performance as MIParT, which is very relevant and oddly omitted for these results since it is shown later on in the case of QG separation.

8: Table 1: Often bold results are used to highlight the best results, not the authors results. That helps the reader to quickly contextualize the new model performance compared to current SOTA. I would use the bold to show the best results for each metric and add “IAFormer (this work)” to the text. Same for Table 2.

9: “Furthermore, the use of sparse attention enables the network to suppress attention scores of less relevant tokens, reducing the need for excessive model complexity, while efficiently distinguishing between top and QCD jets. ” I’m not convinced the reduced variance of the network outputs is necessarily accounted by the sparse attention. For this claim you either need to set beta to 0 to show the case of no sparsity, which is not one of the benchmarks, or possibly implement the sparse attention in the baseline transformer model, to show that in that case the variance of the output is reduced. Moreover, there are additional choices that influence the fluctuations of the output, such as the choice of optimizer, learning rate, convergence criteria, learning rate schedule, so on. While I agree that IAFormer is better than the baselines, I don’t think there’s enough evidence that the sparse attention improves the stability of the outputs.

10: Table 2: The results shown are a bit misleading because the final performance highly depends on the choice of parameterization of the PID, as the authors briefly mentioned. If only experimentally accessible PIDs are used (which I assume is the case for the authors) you get 1/bkg eff at 30% sig eff of around 100 while if you also split the PIDs into all possible categories you get the results where this same metric is ~130. Either the authors split the models between these 2 categories, or evaluate their model based on both options, otherwise the comparison is not fair since the difference of 1% the authors mentioned is only true for metrics like AUC but if you look at PELICAN, it’s better than IAFormer by 50% in the 30%sig eff metric, which again has nothing to do with the network itself. Additionally, the authors omit other models with benchmarks in this dataset such as ABCNet (https://arxiv.org/abs/2001.05311) and PCT (https://arxiv.org/abs/2102.05073), which are also transformer based models.

11: In Fig. 4 how are the particles ordered? I agree that the diagonal has more structure for IAFormer, but the particle ordering is arbitrary, so I do not understand the symmetry argument used (you could swap rows and columns in the 2D maps reuslting in a valid attention map but more messy).

Fig.5: Very interesting plot.

Recommendation

Publish (meets expectations and criteria for this Journal)

---

## Round 1 · Referee Report · Anonymous (Referee 2) · 2025-8-15

Strengths

  • The IAFormer architecture is well motivated and described.
  • IAFormer achieves strong performance at small parameter count on the top-tagging benchmark dataset.
  • The authors apply several interpretability methods to study their architecture.

Weaknesses

  • IAFormer is not benchmarked on the JetClass dataset.
  • The computational cost of IAFormer (timing, FLOPs, memory) is not discussed.
  • The authors do not discuss whether the performance gain comes from differential attention or from using only edge features in attention.

I am concerned that IAFormer (with 200k parameters) will exhibit relatively poor performance on the JetClass dataset, similar to ParticleNet and LorentzNet, and that it will train significantly slower than a plain Transformer due to the memory overhead associated with learnable operations on edge features (i.e., pairwise interaction features). Overall, IAFormer appears closely related to these message-passing graph networks. From my understanding, IAFormer can also be regarded as an extreme variant of MIParT, which only matches ParT’s performance on JetClass when extended to MIParT-L (~2M parameters) while still training substantially slower because of the additional memory overhead. It would be valuable if the authors could address these concerns with further experiments, as outlined in the “Requested changes” section.

Report

The authors present IAFormer, a novel Transformer architecture that matches the performance of Lorentz-equivariant architectures on the top-tagging dataset. However, I suspect that IAFormer is substantially more computationally expensive than ParT at an equal parameter count, and therefore does not scale well to large datasets such as JetClass. The scalability and performance on large-scale datasets are the key questions in practice; it would be great if the authors could clarify them to complete the picture of IAFormer.

Requested changes

Major points:

  1. The main motivation for using transformers for jet tagging is that they scale well to large datasets while keeping manageable computational cost even for millions of parameters. The authors benchmark IAFormer only on the small-scale top tagging and quark-gluon datasets, where message-passing graph networks and transformers achieve similar performance to transformers. To demonstrate that IAFormer is useful for large-scale datasets, the authors should evaluate IAFormer on the JetClass dataset, and perhaps also a IAFormer-L with ~2M parameters (similar to MIParT-L).

  2. The authors write in the abstract "Despite being computationally efficient by more than an order of magnitude than the Particle Transformer network" and probably mean that IAFormer has 10x less parameters than ParT. While small parameter count makes interpretability easier and avoids overfitting, it has little to do with computational efficiency. For instance, a transformer with 2M parameters typically trains faster and uses less memory than a 200k parameter fully-connected graph network. To complete the picture, the authors should compare FLOPs and memory consumption of their plain transformer and IAFormer, and ideally also compare the timing on the same GPU.

  3. IAFormer combines two ingredients, (a) differential attention and (b) using only edge features in attention. The two features do not rely on each other, raising the question of which aspect gives the performance gain. The authors already include a plain transformer baseline, making it natural to also include the seperate cases of only (a) and only (b) in addition to the full IAFormer (a)+(b) for one dataset, e.g. top-tagging. To increase the value of this ablation, the authors should optimize the training hyperparameters and regularization of the plain Transformer, they can e.g. use the ParT training hyperparameters. With a good training, the plain Transformer should achieve AUC values in the range 0.9840 - 0.9850. This study would be very valuable for the community, as differential attention is be easy to implement also in other transformer architectures in HEP. But this is less relevant than the two points above.

Minor points: 1. The three AUCs reported in figure 2 and figure 3 are significantly worse than the AUCs reported in table 1 and table 2. I think figure 3 should have 'accuracy = ...' in the legend, not 'AUC = ...'. I think these just have to be updated with the final results. 2. In table 1 and table 2, the authors list OmniLearn, a fine-tuned network, together with networks trained from scratch. If the authors want to include OmniLearn, they show it next to ParT-f.t., L-GATr-f.t. etc, and seperate it visually from networks trained from scratch. 3. The authors mention that they have to clip the beta in differential attention to the range [0,1]. Hard clipping can lead to unstable training if it is often triggered, a better solution would be to defined beta=sigmoid(gamma) with gamma an unconstrained learnable parameter. Also, how was this handled in the original paper on differential attention? 4. Concerning table 1: The authors only report 3 digits for the AUC of their Plain Transformer and IAFormer, they should report 4. Also, L-GATr has 1.1M parameters, see appendix C.2 in https://arxiv.org/abs/2405.14806. 5. The sentence "Since the interaction matrix is independently added to each Transformer layer." in the paragraph before Section 2.3 is missing something. 6. The authors write in Section 2.3.2 "Unlike original Transformer-based models, our approach does not require a class token to aggregate learned information across the layers.". The class token is a specific trick first used with vision transformers (https://arxiv.org/abs/2103.17239), it is not the standard and also not necessary. Many transformers in HEP use average pooling like IAFormer, e.g. Parnassus and L-GATr. Please modify this sentence. 7. In Table 2, the authors show the ParT and MIParT results in italic letters, because they are trained on additional PID information (see table 2 of the ParT paper https://arxiv.org/abs/2202.03772 for an explanation on the difference between QG_exp and QG_full). For all other networks, the authors report the QG_exp results. The ParT paper also reports the QG_exp result for ParT in table 6, however I did not find a QG_exp result for MIParT in the MIParT paper. It seems that the literature is not consistent on whether to use the PID information in the training. To be consistent, I recommend to show the ParT_exp value from the ParT paper instead of the currently used ParT_full value, and remove the MIParT value. 8. The manuscript contains 79x 'IAFormer' and 3x 'IAformer', please unify.

Recommendation

Ask for major revision

---

## Round 2 · Referee Report · Anonymous (Referee 1) · 2025-12-23

Report

The authors addressed all the points raised during the review and I'm happy to support the publication of the current draft. The only major modification I propose is for the authors to separate the paper pdf from their answers since at first I almost missed the authors replies.

Requested changes

Separate the paper draft from their answers the the referees.

Recommendation

Publish (meets expectations and criteria for this Journal)

---

## Round 2 · Referee Report · Anonymous (Referee 2) · 2026-1-8

Disclosure of Generative AI use

The referee discloses that the following generative AI tools have been used in the preparation of this report:

Used ChatGPT 5.2 to check that I did not overlook any differences between the updated version and the original one.

Report

The authors have submitted a revised manuscript that addresses several of the reviewers comments and adds substantial new material. In particular, the revision introduces a new Section 3.4 presenting JetClass results (IAFormer trained on 10M jets), and restructures the ablation and design-justification studies into a dedicated Section 3.2.1, including a discussion of computational cost. These additions materially improve the manuscript and strengthen the empirical support for the proposed architecture.

The three main concerns raised in my initial report have been addressed, but in a way that remains partially incomplete and leaves a few questions unresolved. I therefore recommend minor revision, and I outline below concrete, targeted changes that would improve clarity and completeness. Most of these should be straightforward to implement. The only potentially time-consuming item is training IAFormer on the full 100M-jet JetClass dataset rather than the 10M-jet proxy.

Finally, I note that many of the minor points from my first report do not appear to be reflected in the resubmission, nor are they discussed in the authors’ response. In addition, the response refers to "the referee", although two referee reports were submitted. This raises the possibility that the authors may not have received my original report.

Requested changes

Major points:

1) JetClass

The authors added section 3.4 on the JetClass dataset, including results for a IAFormer trained on 10M jets, which matches the ParT and MIParT-L literature results while using less parameters. That's interesting to see, thanks for including the study! Comments:

  • I did not find the batchsize and number of epochs used for the JetClass trainings in the paper. Did the authors use the same number of epochs as for the top tagging dataset? E.g. ParT is trained for 5 epochs on the 100M dataset with batch size 512, but MIParT-L is trained for 50 epochs with batch size 384, a significantly higher compute cost.

  • The reported results are for 10M jets, but the JetClass training set contains 100M jets. Why did the authors train on only 10M jets? Please explain, or ideally train on the standard 100M jets.

  • More baselines: Currently only ParT and MIParT-L results are shown, please include also ParticleNet, L-GATr (also 10M results in the papers), and LLoCa-Transformer, LorentzNet [1,2] if trained on the full 100M jets.

2) Computational cost

The authors added a new section 3.2.1 on the comparison with simpler architectures and the computational cost, I think this significantly increases the quality of the paper. In particular, they discuss timing and FLOPs, with a 10x decrease in FLOPs. Comments:

  • Please also report memory usage. Like FLOPs, memory usage is hardware independent, making it a more suitable for comparison. In pytorch, simply extract torch.cuda.max_memory_allocated().

  • Timings are only reported for IAFormer and IAFormer(beta=0). To quantify the overhead of IAFormer, the authors should also report timings for their plain transformer and their ParT-equivalent. Also, the reported 11 seconds per batch is huge, e.g. [1,2] find that 1M JetClass iterations with batchsize 512 take 15h for a plain transformer on a H100 GPU, or 0.05s per iteration, and 0.12s for ParT.

3) Impact of edge features vs differential attention

The new section 3.2.1 and the extended Figure 2 give additional information on this central aspect of the paper. Comments:

  • Please double-check that the 'plain Transformer' and 'Transformer + I_ij' lines are correct. Table 1 reports a rejection rate for the plain transformer that matches the 'Transformer + I_ij' result in Figure 2.

  • If time allows, it would be valuable to add the case 'Transformer + beta' to quantify the impact of differential attention on a plain transformer tagger.

Minor points, remaining from the first report:

1) The mean of the AUCs and rejection rates in figure 2 and 3 seem to not agree with the results reported in table 1 and 2. For instance, in figure 2 (right) the IAFormer rejection rates are always matching or below 2000, but table 1 reports 2012. For the Plain transformer, figure 2 displays ~500 for the rejection rate, but table 1 says 1350. Figure 3 has mean(ACC)=0.843, but table 2 reports 0.844. Please show consistent results.

2) In table 1 and 2, the authors list OmniLearn, a fine-tuned network, next to networks trained from scratch. This is an unfair comparison, because the fine-tuned network is trained on more information. If the authors want to show fine-tuned networks, they should also include others like ParT-f.t., ParticleNet-f.t., L-GATr-f.t., and seperate them with a bar similar to the 'Lorentz invariance based networks'.

3) In Table 1 the authors only report 3 digits for the IAFormer AUC, they should report 4 digits, potentially with uncertainty if not negligible. Figure 2 (left) even reports 5 digits.

4) In Table 1, the L-GATr top-tagger should have 1.1M parameters, not 1.8M, see appendix C.2 in Ref. [3].

5) The authors write in Section 2.3.2 "Unlike original Transformer-based models, our approach does not require a class token to aggregate learned information across the layers.". The class token is a specific trick first used with vision transformers [4], it is not the standard and also not required. Many transformers in HEP use average pooling like IAFormer, e.g. Parnassus and L-GATr.

6) The caption of table 2 should be modified to clearly explain the difference between 'exp' and 'full' trainings. It might help to add 'exp' to all other networks, and refer to the ParT paper for more details.

Extra minor points noticed while studying the resubmission:

1) The authors do not mention how many independent trainings they used to estimate uncertainties in table 1 and 2, this should be part of the caption.

2) The authors should report mean+uncertainty of the bands in Figure 2 (right) to allow comparison with other results from table 1. Additionally, this figure could be more meaningfully displayed as bands with uncertainty, or directly in the style of table 1, including also the other tagging metrics.

[1] https://arxiv.org/abs/2505.20280 [2] https://arxiv.org/pdf/2508.14898 [3] https://arxiv.org/abs/2405.14806 [4] https://arxiv.org/abs/2103.17239

Recommendation

Ask for minor revision

---

## Round 2 · Author Response

The authors thank the referee for reading our manuscript and providing valuable comments. We have
made the appropriate changes based on the comments of the referee. We have addressed all the points
raised by the referee and believe that our manuscript is now ready for publication.

With best regards,
The authors

---

## Editorial Decision

awaiting_resubmission